microbiology, ecology

experimental evolution, evolve-and-resequence, species turnover, gut microbiota, microecology

**Author for correspondence:**
Christian Schlötterer
e-mail: christian.schloetterer@vetmeduni.ac.at

# Long-term gut microbiome dynamics in *Drosophila melanogaster* reveal environment-specific associations between bacterial taxa at the family level

Rupert Mazzucco and Christian Schlötterer

Institut für Populationsgenetik, Veterinärmedizinische Universität Wien, Veterinärplatz 1, Wien 1210, Austria

RM, 0000-0002-7608-0196

The influence of the microbiome on its host is well-documented, but the interplay of its members is not yet well-understood. Even for simple microbiomes, the interaction among members of the microbiome is difficult to study. Longitudinal studies provide a promising approach to studying such interactions through the temporal covariation of different taxonomic units. By contrast to most longitudinal studies, which span only a single host generation, we here present a *post hoc* analysis of a whole-genome dataset of 81 samples that follows microbiome composition for up to 180 host generations, which cover nearly 10 years. The microbiome diversity remained rather stable in replicated *Drosophila melanogaster* populations exposed to two different temperature regimes. The composition changed, however, systematically across replicates of the two temperature regimes. Significant associations between families, mostly specific to one temperature regime, indicate functional interdependence of different microbiome components. These associations also involve moderately abundant families, which emphasizes their functional importance, and highlights the importance of looking beyond the common constituents of the *Drosophila* microbiome.

## 1. Introduction

The interaction of the microbiome with its host is highly topical and has been investigated in many species [1–4]. In particular in the context of human health [5], a wide variety of host conditions has been linked to the microbiome [6–13]. The causal relations between microbiome composition and its effect on the host remain poorly understood [14,15].

The composition of the microbiome is highly dynamic and affected by the host genotype as well as by environmental factors [16–21]. Compositional changes of the microbiome, which are triggered by environmental challenges, provide the potential to indirectly modulate the response of the host to an altered environment [22]. Nevertheless, despite a well-documented turnover of microbiome composition and complexity, our understanding of how microbial communities establish and persist in interaction with their host, and the functional interaction among the members of the microbiome community, is still in its infancy [23,24].

Two approaches to study the functional interactions have been proposed: experimental manipulation of the microbiome composition combined with phenotypic analysis of the host [25–27], and covariance in abundance of specific taxonomic units [28]. The underlying idea is that species with functional independence will vary in their abundance randomly with respect to environmental conditions or host genotypes [29]. Covariation, in contrast, suggests that they are either affected in the same way (positive correlation) or they complement each other (negative correlation). The correlation analysis is, nevertheless,

challenged by spurious associations [30,31], in particular when only one or a few host generations are studied.

By contrast to mammalian microbiomes, the gut microbiome of the fruit fly *Drosophila melanogaster* is relatively simple [32,33]. This provides the opportunity to evaluate the impact of the presence of different combinations of the microbiome on the *Drosophila* host [34,35]. Nevertheless, despite the relatively low complexity of the *Drosophila* microbiome, testing the number of possible combinations is challenging; hence, most studies focus on the most abundant species, (e.g. [36,37]), which in laboratory strains typically belong to the genera *Acetobacter* and *Lactobacillus* [38].

For an unbiased exploration of functional associations of different microbiome components, however, the focus on a pre-selected set of taxa is not well-suited and a covariation approach holds more promise: surveying a large number of individuals or following the abundance dynamics in a longitudinal study, can uncover functionally associated taxa. A challenge for the interpretation of longitudinal covariation patterns is that the starting condition (species composition and abundance pattern) affects subsequent time points [39]. Hence, unless the stochastic fluctuations are large, it can be difficult to disentangle random associations from functional ones, in particular when longitudinal studies are only conducted within a single host generation.

*Drosophila* provides multiple advantages to study functional associations between components of the microbiome. Firstly, the microbiome is not very complex, making the analyses more powerful; secondly, the *Drosophila* microbiome is highly dynamic with large changes in microbiome composition [40], which will break random associations and retain only functionally relevant ones; thirdly, longitudinal studies in mammals often only look at changes within a single host generation, (e.g. [41]). The comparatively short generation time of *Drosophila* enables studies of microbiome time series [39,42] that cover many generations, which provides a substantial increase in power to detect functionally important associations.

Several studies analysed the interplay of thermal adaptation of the *Drosophila* host and its microbiome [43–47], a question of interest in the context of climate change and the possibility of associated range shifts [48,49]. The results remained nevertheless inconclusive, which may be attributed to the small number of host generations covered in those studies. To obtain more robust results, we used an extended longitudinal study of the *D. melanogaster* microbiome during adaptation to novel hot and cold temperature regimes over nearly 10 years, covering 180 generations in hot and 100 generations in cold environments. We characterize the microbiome diversity and changes in composition over time, and analyse associations between taxa and global trends common across replicates, highlighting the significant differences between the two temperature regimes.

## 2. Material and methods

The Pool-Seq dataset (PRJEB37761) used in this study comes from a long-term evolve-and-resequence experiment of 180 generations of *D. melanogaster* kept in hot conditions (temperature fluctuating between 18°C and 28°C to mimic day and night), and 100 generations kept in cold conditions (fluctuating between 10°C and 20°C), with five replicates each, comprising 81 samples in total (electronic supplementary material, table S1).

### (a) Culture conditions

In each temperature regime replicate populations of approximately 1000 flies in five bottles (approx. 200 × 5) were maintained in parallel using the same conditions and sample handling (including sample freezing). The developmental rates differ between temperature regimes, thus the samples in the two different temperature regimes were handled independently. All flies were maintained on a standard fly food medium (agar–agar, sugar beets syrup, malt syrup, yeast, corn flour, soy flour, in weight proportions of about $1:3:3:3:7:1$), which remained the same throughout the entire experiment. Eclosed flies were transferred to a fresh medium for 4 (hot) or 8 (cold) days and upon transfer to the bottles giving rise to the next generations they were supplemented with additional flies, which emerged during this period. After 2 (hot) and 4 (cold) days of egg laying the flies were transferred to another set of fresh bottles for another round of egg laying. After the last round of egg laying the flies were snap frozen in liquid nitrogen and stored at –80°C until DNA extraction.

### (b) DNA extraction and sequencing

DNA was extracted from approximately 500 female flies (except for a single sample that was made from a mixture of males and females; see electronic supplementary material, table S1) using the same high salt extraction procedure for all samples considered here [50]. All reagents used for DNA extraction were prepared in large batches which were used to extract DNA from multiple time points. The exact assignment of batches to specific DNA extractions is not possible. Within a single time point the age of the flies varied between 4 and 8 days for the hot environment and 9–16 days for the cold environment. Pools were sequenced at various time points, using a range of library kits, insert sizes, and read lengths, reflecting the development of Illumina sequencing over a decade ([51]; electronic supplementary material, table S1 and figure S6). Reads were further processed as described below.

### (c) Removing barcoded spike-ins

Six lanes (r01F23.93-96, r05F15.58, r10F15.90) contained some additional spike-in reads from *Aviadenovirus A* and/or *Cochliomyia hominivorax* with separate barcodes. The five-base barcodes PEMx1–4 are found at the 5′ end of each read of a pair. We removed all read pairs with the barcode on both reads, allowing up to one mismatch each, with homebrew R code [52] using functions from Bioconductor::ShortRead v. 1.42.0 [53].

### (d) Quality control and trimming

Quality control with FastQC v. 0.11.8 [54] revealed consistent anomalies with base frequencies over the first few bases on the 5′ end, as well as occasional minor adapter contamination. We, therefore, clipped the first five bases on the 5′ end and removed adapter fragments with trimmomatic v. 0.39 [55] (parameters: ILLUMINACLIP:{adapters}:2:30:10:2 HEADCROP:5 MINLEN:50 AVGQUAL:28, with {adapters} chosen as TruSeq2.fa or TruSeq-3.fa as applicable to the respective library), also discarding reads below a minimum length of 50 and an average quality below 28.

### (e) Contaminant and duplicate removal

Contaminants were removed by mapping the trimmed read pairs with bowtie2 v. 2.3.5.1 [56] against the *D. melanogaster* host and a set of other genomes that were sequenced together with the target libraries, but may have not been fully removed: we used a combined reference of *D. melanogaster* (GCF_000001215.4), *w*Mel (NC_002978.6), *Homo sapiens* (GCF_000001405.39), *Mus musculus* (GCF_000001635.26), *Gallus gallus* (GCF_000002315.4),

**Table 1.** The seven most abundant families across all samples (read counts larger than 1% of the total).

| family | read pairs | fraction | genera |
|---|---|---|---|
| Acetobacteraceae | 12 585 915 | 0.68 | 93% *Acetobacter*, 4% *Komagataeibacter*, 2% *Gluconacetobacter*, 1% *Gluconobacter* |
| Lactobacillaceae | 1 556 816 | 0.084 | 100% *Lactobacillus* |
| Streptomycetaceae | 603 057 | 0.033 | 62% *Streptomyces* |
| Enterobacteriaceae | 397 342 | 0.021 | 27% *Salmonella*, 17% *Escherichia*, 4% *Klebsiella*, 4% *Citrobacter*, 3% *Enterobacter* |
| Burkholderiaceae | 275 562 | 0.015 | 89% *Ralstonia*, 6% *Cupriavidus*, 2.5% *Paraburkholderia*, 2.5% *Burkholderia* |
| Leuconostocaceae | 263 390 | 0.014 | 100% *Leuconostoc* |
| Bradyrhizobiaceae | 195 634 | 0.011 | 100% *Bradyrhizobium* |

*Aviadenovirus A* (NC_001720.1), and Illumina PhiX (modified NC_001422.1 with 587:G > A, 833:G > A, 2731:A > G, 2793:C > T, 2811:C > T), and removed all concordantly mapped pairs (parameters: --end-to-end --maxins 1000 --no-mixed --no-discordant). Although *Wolbachia* is part of the microbiome in a wider sense as it lives inside the host cells, we did not include it in our analysis as is not part of the gut microbiome; its dynamics were studied separately [51,57]. As a final preparation step, duplicates were removed with fastuniq v. 1.1 ([58]; default parameters).

### (f) Classification with Kraken2

Deduplicated read pairs were classified with kraken2 v. 2.0.8-beta [59] (parameters: --paired --confidence 0.04). Kraken2 has two parameters that influence classification. We did not modify the minimum base quality parameter from its default value of 0 as an exploratory analysis confirmed that our stringent quality filtering makes this parameter uninformative. Increasing the confidence parameter did, however, remove rare dubious classifications (i.e. hits to eukaryotes that could not have contaminated the libraries, e.g. golden eagles, salmon, turtles, wine, cucumbers etc. and which were based only on very few k-mers), while also decreasing the number of classified reads. After some exploration we chose $c = 0.04$ for this parameter, as with higher values the disadvantage of losing read pairs outweighed the gain in classification confidence.

We classified read pairs against a database built with the kraken2-build script (default parameters) from the NCBI NT database [60], with higher animals (Metazoa) and plants (Viridiplantae) removed and the dmel6_iso1MT (GCF_000001215.4) RefSeq [61] genome added.

### (g) Postprocessing with bracken

Kraken2 assigns reads on the lowest taxonomic level for which a unique assignment is possible, which implies that different reads can be assigned at different taxonomic levels. We, therefore, used the companion program bracken v. 2.5.0 [62] for re-estimating read counts for a given taxonomic level using a Bayesian model, using the untrimmed read length of each sequencing lane.

We analysed the results on the family level after excluding any remnants of the known contaminants Drosophilidae, Adenoviridae and Anaplasmataceae. Most of the large families are dominated by a single genus, e.g. *Acetobacter*, *Lactobacillus*, *Leuconostoc*, *Ralstonia*, *Bradyrhizobium* (table 1).

### (h) Estimating fractions

Because the samples contain only a very small part of the entire microbiome, observed read counts may be subject to substantial sampling error. Although this issue is often ignored [63], accounting for imperfect detection generally improves the analysis [64,65]. Therefore, given the $m = 806$ families classified across all samples, we estimated their fractions $x_i$, with $i = 1, \ldots, m$, in each sample from their read counts $n_i$ as $x_i = (n_i + 1)/(N + m)$ with $N = \sum n_i$, which assumes that the combined process of library preparation, sequencing, and read extraction is described by a multinomial sampling model. We note that our method is a more refined version of the commonly applied procedure of setting counts of 0 to 1 in observation tables: for an observed count $n = 0$ and $m \ll N$, the estimated count would indeed be $Nx = 1/(1 + m/N) \approx 1$.

### (i) Correlation between families

A known problem of compositional data analysis is that component fractions may exhibit spurious negative correlations [66]. We thus used the normalized and log-transformed component fractions (additive log-ratio transform; [67], choosing as common denominator the fraction of the most abundant family Acetobacteraceae) across all samples with the R function Hmisc::rcorr v. 4.4-0 to obtain Spearman's rank correlation coefficients $p$ as a measure of association between families.

### (j) Piece-wise linear models

Figure 2 indicates the presence of abundance peaks at intermediate time points. Since such intermediate peaks cannot be captured by simple linear models, we identified significant trends by fitting segmented linear models using the R segmented package v. 1.1-0 [68,69] with an initial breakpoint located at generation 90 in the hot environment and generation 50 in the cold environment to allow for an intermediate peak. The algorithm moves and possibly drops this breakpoint as it identifies the best-fitting model, thus identifying both linear trends and intermediate peaks or dips in the data.

## 3. Results

We studied the long-term microbiome dynamics of *D. melanogaster* hosts exposed to novel hot (18/28°C) and cold environments (10/20°C). We first considered the changes in composition over time; then, focusing on the dynamics of the seven most abundant families of microbiota, we identified significant associations between several of them. Finally, we identified significant trends of community restructuring characteristic for the respective environmental conditions.

### (a) Microbiome composition

Across all replicates and environments, the microbiome is dominated by seven bacterial families (table 1 and figure 1), most of which were also commonly observed in previous studies of the *Drosophila* microbiome. Like for other *Drosophila*

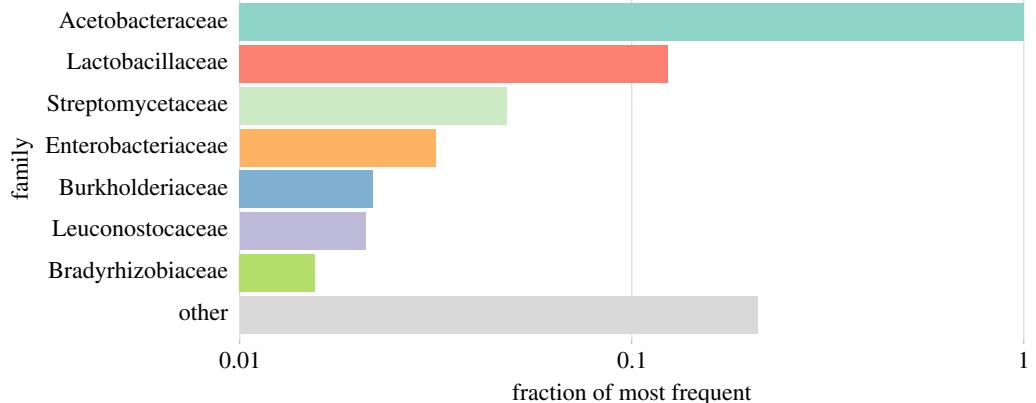

**Figure 1.** Microbial diversity is dominated by seven bacterial families. Analogous to rank-abundance plots, the relative frequencies of reads classified to the most common families are shown. All frequencies are scaled to the frequency of the Acetobacteraceae, the overall most common family (which thus has a value of 1). The colour coding used here applies to all figures throughout the entire manuscript. The seven most abundant families account for at least 74% of all read pairs in each of the 81 samples. (Online version in colour.)

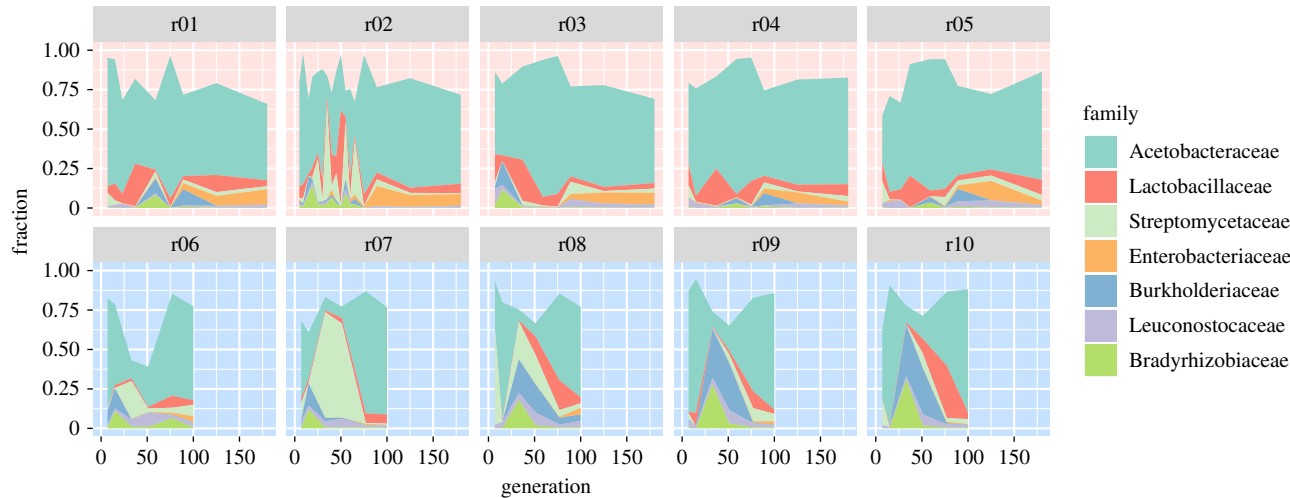

**Figure 2.** Replicate dynamics reveal characteristic features for each environment. Stacked coloured areas indicate the fraction of the respective family in the given generation in each replicate in the hot (upper row, red background) and cold (lower row, blue background) environment, linearly interpolating between sampled generations (9–17 per replicate in the hot environment, 51 in total; 6 per replicate in the cold environment, 30 in total). Only the seven most abundant families (table 1 and figure 1) are shown. (Online version in colour.)

cultured in the laboratory, Acetobacteraceae are the most abundant family [38,44,46], followed by Lactobacillaceae. The predominant families include further families frequently seen in the *Drosophila* microbiome [33], namely, Enterobacteriaceae and Leuconostocaceae. The high abundance of Enterobacteriaceae, a very common *Drosophila*-associated gut microbe family in the wild [43,70,71], is not surprising because the founder isofemale lines were kept only a small number of generations in the laboratory before the experiment was started. Streptomycetaceae are not typically reported as part of the *Drosophila* microbiome, but have been found in a recent study [72]. Finally, the remaining abundant families Bradyrhizobiaceae and Burkholderiaceae are not typical *Drosophila* gut bacteria, but are considered environmental bacteria [73,74].

## (b) Microbiome dynamics

Given that we transferred flies with their associated microbiome to novel environments, we anticipated one of three different scenarios: (i) the microbiome quickly adapts to the new environment, as frequently seen for altered nutrition, (e.g. [75–77]), and then only experiences stochastic changes,

(ii) the microbiome changes continuously because adaptation to the new environment cannot be achieved by a simple change in relative frequency of the community members or (iii) the microbiome remains completely unaffected. None of these predictions fully applied to this experiment; instead, we observed complex dynamics that were also very different between environments (figure 2). Although these changes were not completely synchronized, their occurrence in multiple replicates strongly suggests that these dynamics are not just stochastic effects, but possibly reflect functional turnover of the microbiome during the experiment. In addition to these global patterns, we observed that some bacterial families changed their abundance in a continuous manner; for example, Lactobacillaceae continuously increase in frequency in the cold environment.

## (c) Strong associations

While most studies describe microbiome composition in either natural flies or flies exposed to laboratory conditions, our time series data, covering up to 180 host generations, offer the unprecedented opportunity to test for associations

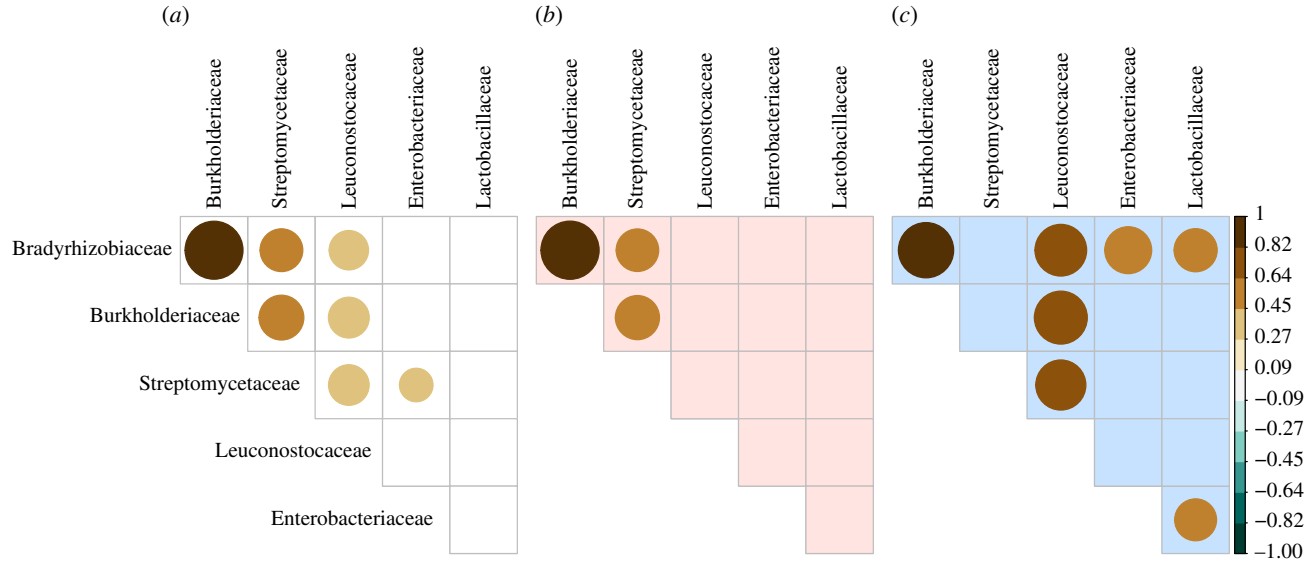

**Figure 3.** Positive associations among families in hot and cold environments. Filled circles indicate Spearman's correlation coefficient between ALR-transformed family fractions over all replicates in (*a*) all, (*b*) only hot and (*c*) only cold environments by size and fill colour. Only associations with *p*-values < 0.01 are shown. Except for the consistent strong association between Burkholderiaceae and Bradyrhizobiaceae, associations between families are different between environments: Streptomycetaceae are associated with this BB group in the hot environment, while in the cold environment Leuconostocaceae have a substantial association with the BB group and the Streptomycetaceae, and the Enterobacteriaceae and Lactobacillaceae are associated with the Bradyrhizobiaceae and each other. (Online version in colour.)

between different components of the microbiome. Given that the microbiome composition is highly stochastic, with substantial changes between generations [78], large abundance changes are expected. Time-series data provide the opportunity to distinguish between stochastic changes and covariation of abundance patterns between microorganisms. Statistically significant associations imply a functional association between covarying taxonomic groups.

As the interaction of members from different families may be informative about changes in the functional composition of the microbiome, we evaluated the correlated dynamics of the predominant families (figure 3). Bradyrhizobiaceae and Burkholderiaceae have strong positive associations across generations and environments (figure 3*a*–*c*). In the hot environment, they are also significantly associated with Streptomycetaceae, while in the cold environment Leuconostocaceae appear to play a pivotal role, being strongly associated with Bradyrhizobiaceae and Burkholderiaceae on the one hand, and Streptomycetaceae on the other, where also Enterobacteriaceae and Lactobacillaceae are associated with Bradyrhizobiaceae and each other. These highly significant associations provide very strong evidence for the interaction between different members of the microbiome, which only become apparent with the availability of long-term longitudinal data.

### (d) Global trends

Because the visual inspection of the abundance plots (figure 2) suggests that the abundance of many families changes in a nonlinear manner, we analysed the family dynamics with segmented linear models to identify significant trends across replicates (figure 4). In cold culture conditions we find clear evidence a strong decrease of Acetobacteraceae and increase of Bradyrhizobiaceae and Burkholderiaceae around generation 30, which is followed by a strong increase of Leuconostocaceae around generation 50. These families, however, then decrease again in relative

abundance as Acetobacteraceae recover and Lactobacillaceae slowly increase (figure 4). The increase of Lactobacillaceae is restricted to the cold environment; in the hot environment, we see the opposite trend—if anything. Interestingly, this pattern of opposing trends in the two environments is seen even more clearly in the Leuconostocaceae, the other member of the group of lactic acid bacteria (electronic supplementary material, §A). Finally, the highly significant increase of Enterobacteriaceae over time occurs in both environments, and thus may be causally unrelated to the other interactions.

## 4. Discussion

Based on a comprehensive longitudinal dataset of 81 samples from replicated populations covering up to 180 host generations, we found strong evidence for a functional interdependence of various families constituting the *Drosophila* microbiome: (i) parallel changes in abundance patterns across replicates and (ii) significant covariation in abundance of several families.

### (a) Parallel dynamics across replicates

The striking compositional similarity of replicate populations in this experiment can be recognized by the clustering of replicates as well as by the similar patterns of frequency changes. The parallel dynamics are most apparent from characteristic, highly pronounced frequency changes of a few families. One example for this is the decrease of Acetobacteraceae at generation 30 across all five replicates in the cold, followed by a recovery at later generations. Interestingly, almost the opposite trend was seen in the hot environment (figure 4).

One explanation for the high parallelism between replicates, which is independent of functional requirements, is related to the culturing method. As the experimental *Drosophila* populations were not maintained under sterile conditions, components of the microbiome may have been transferred accidentally across

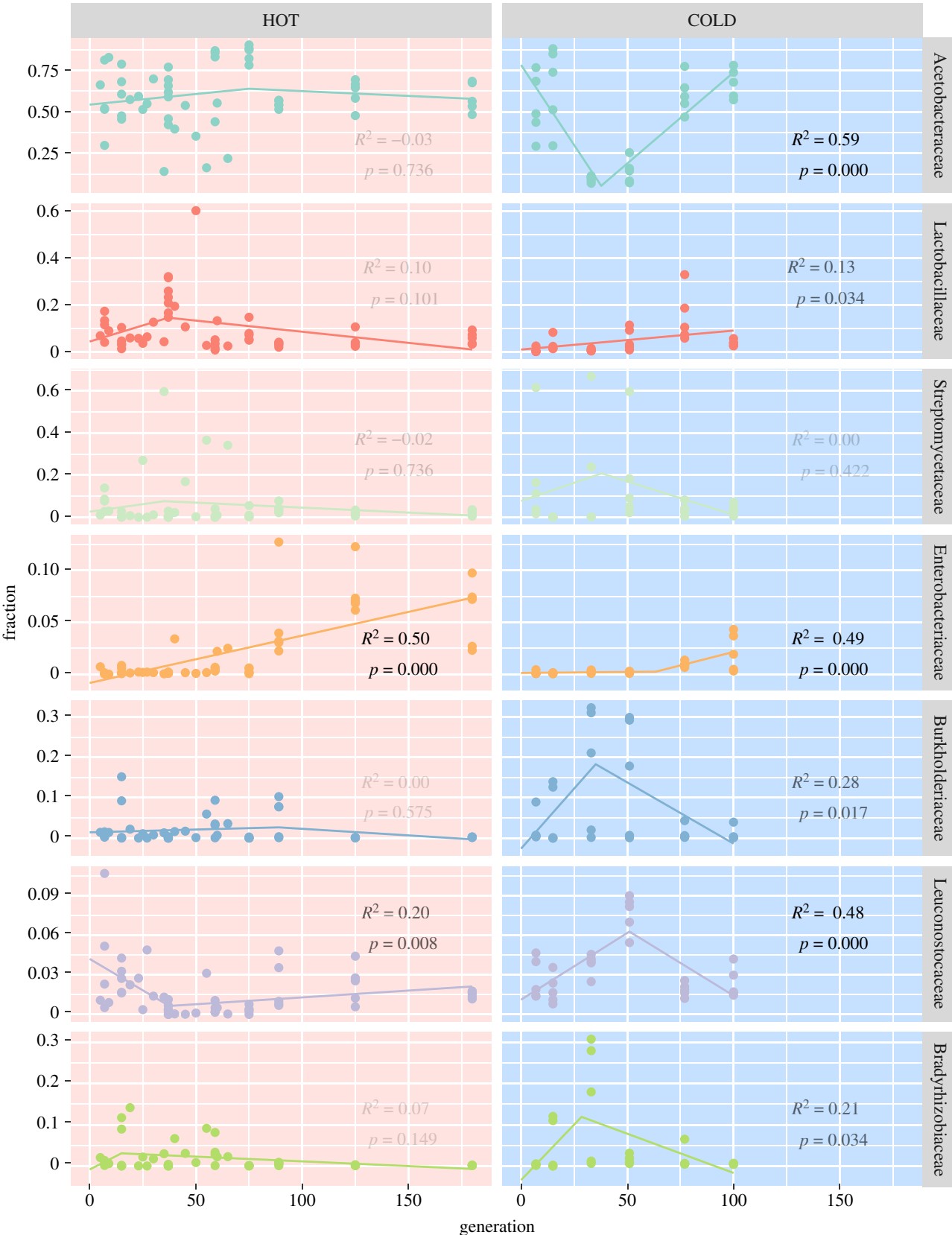

**Figure 4.** Significant trends differ between environments. Coloured points indicate family frequencies in all replicates in the hot (left column, red background) and cold (right column, blue background) environment (colours as in figure 3, families are also indicated on the margins). Lines indicate the best-fitting segmented linear model, with adjusted $R^2$- and $p$-values (Benjamin–Hochberg-corrected for multiple testing and rounded to three digits) given in the respective panel; their opaqueness further indicates significance. The following dynamics were statistically significant: in the hot environment (i) a steady increase of Enterobacteriaceae and (ii) an abundance drop of Leuconostocaceae at generation 37; in the cold environment (iii) a Leuconostocaceae peak around generation 50, which is preceded by (iv) a significant Actetobacteraceae dip at generation 35 and (v) a marginally significant peak in Burkholderiaceae and Bradyrhizobiaceae occurring around the same time; furthermore (vi) a significant increase in Enterobacteriaceae (after generation 60) and (vii) a weakly significant steady increase of Lactobacillaceae. (Online version in colour.)

replicates, thus synchronizing their dynamics. A similar mechanism was recently proposed for the high parallel selection responses of the *Drosophila* host across replicates [79].

Nevertheless, a closer inspection of these dynamics shows that this explanation is unlikely. The diversity peak in the cold culturing regime (figures 3 and 4) is a good example, occurring in a

similar manner across all replicates. Since rare taxa, rather than abundant families, increased in frequency, this is difficult to reconcile with microbiome transfer across replicates: a large increase in frequency by migration requires that many representatives of this family are being transferred. The highly parallel diversity increase makes an extrinsic contamination also unlikely, as all five replicates of the hot temperature regime are affected. Hence, we conclude that accidental contamination during the culturing is unlikely to explain the parallel microbiome dynamics seen in our experiment.

We acknowledge that the use of different sequencing platforms and library preparation protocols [51] could have contributed to the temporal heterogeneity seen in this study (electronic supplementary material, table S2). Importantly, some of the replicates were sequenced with different sequencing platforms (electronic supplementary material, §B and figure S2), and similar pronounced abundance changes can be seen for samples for which the same DNA was sequenced with two different platforms (electronic supplementary material, figure S3a). Furthermore, the temporal dynamics were apparent among time points sequenced using the same technology (electronic supplementary material, figure S3b). While our analyses confirm a certain effect of technology on detected abundance (electronic supplementary material, figures S4 and S5), as previously reported [80–82]; a detailed analysis of the dynamics in each replicate (electronic supplementary material, §C and figures S6 and S7), allows us to conclude that no prominent features of the family dynamics are artefacts of the sequencing technology.

Another contribution to temporal heterogeneity is the degree of gut colonization in the adult flies, which was shown to continue over several days [83]. Since flies were not collected at exactly the same day after eclosure, it is possible that some of the observed changes in abundance may—at least to some extent—reflect heterogeneity in gut colonization. Nevertheless, the functional interdependence of different families inferred from covariation of abundance patterns would be even further strengthened by heterogeneity in colonization status (see below).

## (b) Significant covariation between families

Even stronger evidence for the functional non-independence of co-occurring families in the microbiome comes from the significant correlation in abundance between different families (figure 3). Since this signal of covariation comes from the joint analysis of multiple time points, it cannot be explained by contamination across replicates, which would have been limited to single generations. Furthermore, the large fluctuations in abundance patterns across time points highlights that the significant association between various families reflects strong functional links between them and rules out statistical artefacts. It was particularly striking that most of these associations do not involve components of the *Drosophila* microbiome that were previously used for functional testing, (e.g. [36,84–86]).

## (c) The microbiome turnover: driver or passenger?

A naive expectation for the impact of rearing temperature on microbiome composition is that the microbiome either changes rapidly as it is exposed to a new environment, and persists in the new, temperature-optimized composition; or, that it changes gradually, reflecting the acquisition of new adaptive

mutations in the microbiome. Interestingly, our data do not fit either of these simple expectations. Rather, we noted a highly dynamic nature of the microbiome with trends that are largely consistent across replicates. The parallel dynamics in multiple replicates of the same temperature regime across multiple generations rules out stochastic changes, but raises the question about the underlying cause. While new functional mutations can occur [87], their appearance should be stochastic and not synchronized across replicates. Contingency on a certain genetic background for certain mutations to be successful has been described for citrate use in *Escherichia coli* [88] and a similar principle may apply to the microbiome composition as well. Nevertheless, this would only explain why some changes do not occur early in the experiment, but cannot synchronize the dynamics across replicates.

Here, we propose that much of the non-random patterns in the microbiome are driven by the *Drosophila* host. Several evolve-and-resequence studies exhibit a highly parallel selection response across replicates [89,90]. Given that the host genotype affects the microbiome composition as demonstrated by GWAS [17,18], genetic changes in the host genome, which are parallel across replicates, may also shape the microbiome composition. A direct link between temperature adaptation of the host and microbiome composition has been recently found [91] where microbiome composition, specifically the ratio of acetic acid bacteria to lactic acid bacteria, changes with latitude of *D. melanogaster* populations from eastern North America. While we do not find strong evidence for this antagonism of acetic and lactic acid bacteria between hot- and cold-evolved populations (electronic supplementary material, figure S1), we consider the influence of the host genotype the best explanation for the high parallelism and most likely for much of the temporal dynamics of the microbiome discovered in this study. Our previous analysis of the *Wolbachia* dynamics in this population [51] showed a turnover of different *Wolbachia* genotypes, but these were monotonic and thus unlikely to contribute to the abundance peaks observed in this study.

If the microbiome dynamics are driven by the *Drosophila* host, the continuous turnover of the microbiome in combination with the linear frequency change of Lactobacillaceae in the cold and the Enterobacteriaceae in the hot environment, may imply that the host has not yet fully adapted to the new environment (i.e. has not yet reached the new trait optimum). Nevertheless, it is important to consider that even after reaching the trait optimum the allele frequencies of contributing loci still experience considerable allele frequency changes [92,93], which may drive microbiome changes even when the phenotype of the host does not change anymore after the trait optimum has been reached. Alternatively, some of the dynamics, in particular those not parallel across replicates, may reflect the acquisition of new mutations by the host; (e.g. [94,95]). Single-strain characterization of evolved microbiomes will be helpful to characterize new mutations that occurred during the experiment. In the current set-up, the distinction between ancestral low frequency genotypes and new mutations may be challenging.

## 5. Conclusion

Strong functional associations have been shown before (reviewed in [96]), but these studies typically manipulated

only a small number of taxa to study the influence of the host. In our experiment, we followed the natural dynamics and inferred the patterns of covariation without any artificial manipulation, thus providing a more natural setting than experimentally combining different components of the microbiome. Our analyses indicate that the strongest associations between different components of the microbiome do not involve the most abundant families, but members of intermediate abundance, which are often ignored in functional studies of the *Drosophila* microbiome. Our results highlight the importance of expanding the functional characterization of the *Drosophila* microbiome beyond highly abundant families like Acetobacteraceae and Lactobacillaceae.

Data accessibility. The dataset used in this study is available from the European Nucleotide Archive (PRJEB37761).

Authors' contributions. R.M.: formal analysis, writing—original draft, writing—review and editing; C.S.: conceptualization, funding acquisition, writing—original draft, writing—review and editing. All authors gave final approval for publication and agreed to be held accountable for the work performed therein.

Competing interests. We declare we have no competing interests.

Funding. This work has been supported by the Austrian Science Funds (FWF, P27630) and the European Research Council (ERC, ArchAdapt).

Acknowledgements. Illumina sequencing for a subset of the data was performed at the VBCF NGS Unit (http://www.viennabiocenter.org/facilities). Special thanks to Viola Nolte for preparing and managing sequence data.

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
