## [Peer Review File · Proceedings of the Royal Society B: Biological Sciences]

Review History

RSPB-2021-1203.R0 (Original submission)

Review form: Reviewer 1

Recommendation

Accept with minor revision (please list in comments)

Scientific importance: Is the manuscript an original and important contribution to its field?

Good

General interest: Is the paper of sufficient general interest?

Good

Quality of the paper: Is the overall quality of the paper suitable?

Good

Is the length of the paper justified?

Yes

Should the paper be seen by a specialist statistical reviewer?

No

Do you have any concerns about statistical analyses in this paper? If so, please specify them explicitly in your report.

No

It is a condition of publication that authors make their supporting data, code and materials available - either as supplementary material or hosted in an external repository. Please rate, if applicable, the supporting data on the following criteria.

Is it accessible?

Yes

Is it clear?

Yes

Is it adequate?

Yes

Do you have any ethical concerns with this paper?

No

Comments to the Author

Mazzucco and Schloterer present an analysis of microbiome composition in evolving populations of *D. melanogaster* over a replicated, long term experiment with hot and cold treatments. They find consistent dynamics of the microbiome at the family level across experimental replicates, suggesting metabolic consistency in the microbiome adaptation to the new environment. Furthermore, they find correlations between family level abundances that are consistent across replicates. Overall their results suggest a coarse-grained predictability of microbiome composition in adaptation to novel environments.

I find the work interesting, the analysis complete, and the conclusions for the most part justified. I have only minor comments that I think should be addressed before acceptance of the manuscript.

Main comment: while the analysis is technically solid, there are caveats to the microbiome sequencing and the associations are only correlations. For that reason, I think the conclusions as expressed in the title, abstract, and conclusion sections need to be softened.

Title: line 2, I would remove 'tight' from the title. While some correlations appear tight, they are just correlations, and overall the associations are not what I would call tight from looking at the data. Furthermore it is unclear what these associations might be caused by -- they may be completely indirect. A more justifiable title in my opinion could be 'Long-term gut microbiome dynamics in *Drosophila melanogaster* reveal environment-specific correlations between bacterial taxa at the family level'.

Abstract lines 22-23: I disagree that the evidence is 'unprecedented' or 'striking.' That wording should be toned down.

Introduction

line 55-57: the authors say that there are too many combinations for testing all configurations of *Drosophila* microbiome species, but see Lee 2019; Gould 2018. Both used fully factorial experimental designs with diversity similar to the present work to analyze *Drosophila* microbiomes. I suggest modifying the sentence to reflect this work or removing the sentence.

Methods

DNA extraction to determine microbial community has been shown to be a major source of error and variation in microbiome sequencing experiments. In particular, the extraction methods

produce differential lysis of different types of bacteria, complicating the assignment of relative taxonomic information to different samples in different batches. Over a decade of trial and error by the microbiome community has yielded mechanical lysis as the standard approach. It appears that a standard genomic extraction without mechanical lysis was used. The trends seen in this dataset could thus be influenced by the extraction method. More explanation of the extraction methods and the caveats would resolve this issue.

The age in days after eclosion when the flies were harvested for sequencing is also a potential influence on the bacterial community. According to Blum et al 2013, the fly gut is not substantially colonized until 4 days post eclosion. Changes in colonization occur as the fly ages. Because flies were sampled from 4 to 8 days post eclosion, significant ecological differences in colonization could occur over the initial 4 day period, leading to differences based on sampling. I think the authors should note this caveat.

The authors mention that nutrients influence the microbiome, but what are the nutrients in the fly media used?

What were the temperatures for the hot and cold environments? This information may be in a previous publication, but please provide it here.

A detailed explanation of the culture conditions, including the approximate population sizes, food composition, how frequently the food is changed, etc would help in understanding why seemingly non-fly bacteria persist at high relative abundance in the experiment. For example, *Bradyrhizobia* and *Burkholderia* genomes are presumably environmental bacteria living on the food and not *Drosophila* gut residents. Explaining, based on the sampling methods, how sequencing reads from these strains could be observed at moderate abundances would greatly improve the manuscript.

How many reads were bacterial and informative for each replicate? This could be added as column S in the table S1.

line 277-284. This section is unclear (though the writing in the rest of the paper is clear and logical). Is the idea that this analysis is forthcoming in future work? Examining changes in DNA sequence rather than changes in abundance would not be affected by the DNA extraction methods and would provide an interesting perspective on microbiome evolution. As an aside, the Katie Pollard's lab has developed methods for such inference in human microbiomes, allowing individual strains to be tracked over time. Might those methods be applied here [in future work]?

Discussion

How does the presence of *Wolbachia* correlate with the external microbiome analyzed here?

Various authors have reported that correlations exist or not. Based on the authors' previous work with *Wolbachia* on the same data set, I think the *Wolbachia* dynamics should be related to the present analysis. For instance, could the trends in Clade V account for the shifts in abundance around generation 60?

Since examining bacterial reads from fly sequencing experiments might be repeated by future data mining efforts, some commentary would be helpful on the caveats of the methods and how these can be mitigated.

Review form: Reviewer 2

Recommendation

Major revision is needed (please make suggestions in comments)

Do you have any concerns about statistical analyses in this paper? If so, please specify them explicitly in your report.

Yes

It is a condition of publication that authors make their supporting data, code and materials available - either as supplementary material or hosted in an external repository. Please rate, if applicable, the supporting data on the following criteria.

Is it accessible?

Yes

Is it clear?

Yes

Is it adequate?

No

Comments to the Author

The goal of this work was to use a multigenerational host dataset to examine microbiome changes over a decade in two environmental conditions. Understanding longitudinal change is an important question in the microbiome literature, and few studies have long-term datasets to address it. This study uses up to 180 generations of *Drosophila*, with microbiome sampling of 7 to 16 of the generations. The results, that microbiome diversity and composition change over time and display different patterns in the two temperature regimes, have the potential to be a useful contribution to the microbiome literature. However, I have a few major concerns. I was asked to be a statistical reviewer, so most of my comments will focus on the data analysis.

First, there are major clarity issues in what data were actually analyzed. Sample sizes are not included in the main text. The main text mentions that 100 generations of flies were used from the cold environment and 180 from the hot environment, with 5 replicate populations per time point "obtained from multiple timepoints" (L113). However, supplementary table S1 shows that 10-16 time points per replicate had microbiome data in the hot environment and 7-13 in the cold environment. The main text and figures need to state how many generations were actually *sampled* rather than only how many generations were run; a figure showing sampling design could be useful.

Similarly, it is unclear what results are shown in the figures. For example, the description of Fig. 1 in the text (L227-228) is, "the microbiomes of some consecutive time points were similar and clustered together (Fig. 1)" but the legend for Fig. 1 does not include the generation number or unit of distance on the Y axis so it's not clear how this shows temporal generation differences, or even what generations or replicates are included in the figure. The title for Fig. 2 is "Figure 2 Microbial diversity depends on the environment and changes over time." but nowhere in the legend text or figures is time mentioned. It is unclear what the points represent in figures S4, Fig. 4 includes colors in the scale not present on the figure, and Fig. S1 needs a legend for the color scheme.

The second major issue is how the data were processed. The variation in microbiome results across DNA extraction methods and sequencing technologies is a well-known issue in the microbiome field, and samples need to be processed using consistent methods to be comparable. Because this study combined so many methodologic techniques and these techniques change over time (as the legend for Fig. S6, "Since samples were typically sequenced as they became available, the use of different platforms is correlated with time - which could affect the interpretation of the observed temporal dynamics of the microbiome (Fig. 3) if the sequencing

technology affects species representation.”) it is hard to argue that temporal results represent true changes in the microbiome rather than data processing artifacts. For example, is the unique peak at generation 50 from using a distinct sequencing technology? The paper does a good job acknowledging multi-platform issues (L339 - 347) but it is not clear they controlled for these issues statistically. I do not find figures S6 and S7 sufficiently convincing that the results can be explained by time and not technology.

The third issue is clarity in the statistical results. Piecewise linear models are not commonly used in microbiome studies, so a sentence justifying their use would be helpful. It also would be useful to list what variables were included in the models- e.g. were replicate, DNA extraction method, and sequencing platform controlled for? The statistical results are often reported in qualitative terms, with no test statistics or other numbers shown. For example, L168 states that “analyses at the genus level provided a very similar picture” but these results are not shown; L231 - 233 describes values as ‘statistically more diverse’ with no stats; L242-243 lists the most abundant families but not their percentage abundances; and L270-L273 describes values ‘highly variable between timepoints’. Is there supplementary info that I’m missing that includes the actual statistical model results? The SI info I have only shows figures.

There are also a few minor points that would stand to improve the paper, bulleted below:

- o L172-L177 Please add a reference for using estimated instead of observed fractions.
- o L205 - L217- This paragraph should be removed or altered- it reads like an abstract rather than as part of a results section.
- o Fig. 4- Is there a quantitative analysis on the ‘obvious groups’?
- o Fig. 5- The grey frames are very hard to see.

Decision letter (RSPB-2021-1203.R0)

13-Aug-2021

Dear Dr Mazzucco:

I am writing to inform you that your manuscript RSPB-2021-1203 entitled "Long-term gut microbiome dynamics in *Drosophila melanogaster* reveal tight, environment-specific associations between specific families" has, in its current form, been rejected for publication in Proceedings B.

This action has been taken on the advice of referees, who have recommended that substantial revisions are necessary. With this in mind we would be happy to consider a resubmission, provided the comments of the referees are fully addressed. However please note that this is not a provisional acceptance.

- 1) A ‘response to referees’ document including details of how you have responded to the comments, and the adjustments you have made.

- 2) A clean copy of the manuscript and one with 'tracked changes' indicating your 'response to referees' comments document.
- 3) Line numbers in your main document.
- 4) Data - please see our policies on data sharing to ensure that you are complying (<https://royalsociety.org/journals/authors/author-guidelines/#data>).

Sincerely,
Dr Sasha Dall
mailto: proceedingsb@royalsociety.org

Associate Editor

Comments to Author:

Both reviewers recognized the value of this paper as it comprises a robust longitudinal dataset of microbiome changes in *Drosophila*. However, the reviewers also pointed out some inconsistencies related to sample sizes, suggested new figures to better explain the sampling design, and highlighted potentially confounding factors in the sequencing approach.

Reviewer(s)' Comments to Author:

Referee: 1

Comments to the Author(s)

Mazzucco and Schloterer present an analysis of microbiome composition in evolving populations of *D. melanogaster* over a replicated, long term experiment with hot and cold treatments. They find consistent dynamics of the microbiome at the family level across experimental replicates, suggesting metabolic consistency in the microbiome adaptation to the new environment. Furthermore, they find correlations between family level abundances that are consistent across replicates. Overall their results suggest a coarse-grained predictability of microbiome composition in adaptation to novel environments.

I find the work interesting, the analysis complete, and the conclusions for the most part justified. I have only minor comments that I think should be addressed before acceptance of the manuscript.

Main comment: while the analysis is technically solid, there are caveats to the microbiome sequencing and the associations are only correlations. For that reason, I think the conclusions as expressed in the title, abstract, and conclusion sections need to be softened.

Title: line 2, I would remove 'tight' from the title. While some correlations appear tight, they are just correlations, and overall the associations are not what I would call tight from looking at the data. Furthermore it is unclear what these associations might be caused by -- they may be completely indirect. A more justifiable title in my opinion could be 'Long-term gut microbiome dynamics in *Drosophila melanogaster* reveal environment-specific correlations between bacterial taxa at the family level'.

Abstract lines 22-23: I disagree that the evidence is 'unprecedented' or 'striking.' That wording should be toned down.

Introduction

line 55-57: the authors say that there are too many combinations for testing all configurations of *Drosophila* microbiome species, but see Lee 2019; Gould 2018. Both used fully factorial experimental designs with diversity similar to the present work to analyze *Drosophila* microbiomes. I suggest modifying the sentence to reflect this work or removing the sentence.

Methods

DNA extraction to determine microbial community has been shown to be a major source of error and variation in microbiome sequencing experiments. In particular, the extraction methods produce differential lysis of different types of bacteria, complicating the assignment of relative taxonomic information to different samples in different batches. Over a decade of trial and error by the microbiome community has yielded mechanical lysis as the standard approach. It appears that a standard genomic extraction without mechanical lysis was used. The trends seen in this dataset could thus be influenced by the extraction method. More explanation of the extraction methods and the caveats would resolve this issue.

The age in days after eclosion when the flies were harvested for sequencing is also a potential influence on the bacterial community. According to Blum et al 2013, the fly gut is not substantially colonized until 4 days post eclosion. Changes in colonization occur as the fly ages. Because flies were sampled from 4 to 8 days post eclosion, significant ecological differences in colonization could occur over the initial 4 day period, leading to differences based on sampling. I think the authors should note this caveat.

The authors mention that nutrients influence the microbiome, but what are the nutrients in the fly media used?

What were the temperatures for the hot and cold environments? This information may be in a previous publication, but please provide it here.

A detailed explanation of the culture conditions, including the approximate population sizes, food composition, how frequently the food is changed, etc would help in understanding why seemingly non-fly bacteria persist at high relative abundance in the experiment. For example, *Bradyrhizobia* and *Burkholderia* genomes are presumably environmental bacteria living on the food and not *Drosophila* gut residents. Explaining, based on the sampling methods, how sequencing reads from these strains could be observed at moderate abundances would greatly improve the manuscript.

How many reads were bacterial and informative for each replicate? This could be added as column S in the table S1.

line 277-284. This section is unclear (though the writing in the rest of the paper is clear and logical). Is the idea that this analysis is forthcoming in future work? Examining changes in DNA sequence rather than changes in abundance would not be affected by the DNA extraction methods and would provide an interesting perspective on microbiome evolution.

As an aside, the Katie Pollard's lab has developed methods for such inference in human microbiomes, allowing individual strains to be tracked over time. Might those methods be applied here [in future work]?

Discussion

How does the presence of *Wolbachia* correlate with the external microbiome analyzed here?

Various authors have reported that correlations exist or not. Based on the authors' previous work with *Wolbachia* on the same data set, I think the *Wolbachia* dynamics should be related to the present analysis. For instance, could the trends in Clade V account for the shifts in abundance around generation 60?

Since examining bacterial reads from fly sequencing experiments might be repeated by future data mining efforts, some commentary would be helpful on the caveats of the methods and how these can be mitigated.

Referee: 2

Comments to the Author(s)

The goal of this work was to use a multigenerational host dataset to examine microbiome changes over a decade in two environmental conditions. Understanding longitudinal change is an important question in the microbiome literature, and few studies have long-term datasets to address it. This study uses up to 180 generations of *Drosophila*, with microbiome sampling of 7 to 16 of the generations. The results, that microbiome diversity and composition change over time and display different patterns in the two temperature regimes, have the potential to be a useful contribution to the microbiome literature. However, I have a few major concerns. I was asked to be a statistical reviewer, so most of my comments will focus on the data analysis.

First, there are major clarity issues in what data were actually analyzed. Sample sizes are not included in the main text. The main text mentions that 100 generations of flies were used from the cold environment and 180 from the hot environment, with 5 replicate populations per time point "obtained from multiple timepoints" (L113). However, supplementary table S1 shows that 10-16 time points per replicate had microbiome data in the hot environment and 7-13 in the cold environment. The main text and figures need to state how many generations were actually *sampled* rather than only how many generations were run; a figure showing sampling design could be useful.

Similarly, it is unclear what results are shown in the figures. For example, the description of Fig. 1 in the text (L227-228) is, "the microbiomes of some consecutive time points were similar and clustered together (Fig. 1)" but the legend for Fig. 1 does not include the generation number or unit of distance on the Y axis so it's not clear how this shows temporal generation differences, or even what generations or replicates are included in the figure. The title for Fig. 2 is "Figure 2 Microbial diversity depends on the environment and changes over time." but nowhere in the legend text or figures is time mentioned. It is unclear what the points represent in figures S4, Fig. 4 includes colors in the scale not present on the figure, and Fig. S1 needs a legend for the color scheme.

The second major issue is how the data were processed. The variation in microbiome results across DNA extraction methods and sequencing technologies is a well-known issue in the microbiome field, and samples need to be processed using consistent methods to be comparable. Because this study combined so many methodologic techniques and these techniques change over time (as the legend for Fig. S6, "Since samples were typically sequenced as they became available, the use of different platforms is correlated with time - which could affect the interpretation of the observed temporal dynamics of the microbiome (Fig. 3) if the sequencing technology affects species representation.") it is hard to argue that temporal results represent true changes in the microbiome rather than data processing artifacts. For example, is the unique peak at generation 50 from using a distinct sequencing technology? The paper does a good job acknowledging multi-platform issues (L339 - 347) but it is not clear they controlled for these issues statistically. I do not find figures S6 and S7 sufficiently convincing that the results can be explained by time and not technology.

The third issue is clarity in the statistical results. Piecewise linear models are not commonly used in microbiome studies, so a sentence justifying their use would be helpful. It also would be useful to list what variables were included in the models- e.g. were replicate, DNA extraction method, and sequencing platform controlled for? The statistical results are often reported in qualitative terms, with no test statistics or other numbers shown. For example, L168 states that "analyses at the genus level provided a very similar picture" but these results are not shown; L231 - 233 describes values as 'statistically more diverse' with no stats; L242-243 lists the most abundant families but not their percentage abundances; and L270-L273 describes values 'highly variable between timepoints'. Is there supplementary info that I'm missing that includes the actual statistical model results? The SI info I have only shows figures.

There are also a few minor points that would stand to improve the paper, bulleted below:

- o L172-L177 Please add a reference for using estimated instead of observed fractions.
- o L205 - L217- This paragraph should be removed or altered- it reads like an abstract rather than as part of a results section.

- o Fig. 4- Is there a quantitative analysis on the 'obvious groups'?
- o Fig. 5- The grey frames are very hard to see.

Author's Response to Decision Letter for (RSPB-2021-1203.R0)

See Appendix A.

RSPB-2021-2193.R0

Review form: Reviewer 2

Recommendation

Accept with minor revision (please list in comments)

Do you have any concerns about statistical analyses in this paper? If so, please specify them explicitly in your report.

No

It is a condition of publication that authors make their supporting data, code and materials available - either as supplementary material or hosted in an external repository. Please rate, if applicable, the supporting data on the following criteria.

Is it accessible?

Yes

Is it clear?

Yes

Is it adequate?

Yes

Comments to the Author

I appreciate how much work the authors have put into revising the manuscript, especially the addition of the code files, supplementary text, and new supplemental figures (S6 and S7). I think in particular they have done a solid job both in the text and response to reviewers arguing that their results are not due to technical issues, and better explaining their study design. I also appreciate their thorough responses to my critiques about statistical approaches and sequencing platforms. I only have a few minor comments.

- o Figure 1. It would be useful to add a sentence explaining that the additive log ratio uses Acetobacteraceae as the denominator, hence its value as 1 and other taxa are scaled relative to Acetobacteraceae. This addition would make the figure easier to interpret without reading the main text.
- o Figure 3 needs a legend.
- o The supplementary text states 'these cannot be from sequencing platforms', but I suggest rephrasing that 'these are highly unlikely to be from sequencing platforms'.

Decision letter (RSPB-2021-2193.R0)

19-Nov-2021

Dear Dr Mazzucco

I am pleased to inform you that your manuscript RSPB-2021-2193 entitled "Long-term gut microbiome dynamics in *Drosophila melanogaster* reveal environment-specific associations between bacterial taxa at the family level" has been accepted for publication in Proceedings B.

The referee(s) have recommended publication, but also suggest some minor revisions to your manuscript. Therefore, I invite you to respond to the referee(s)' comments and revise your manuscript. Because the schedule for publication is very tight, it is a condition of publication that you submit the revised version of your manuscript within 7 days. If you do not think you will be able to meet this date please let us know.

Sincerely,

Dr Sasha Dall

Associate Editor

Board Member

Comments to Author:

The Reviewer was pleased with the improvements in the manuscript. However, also pointed out to some small changes in the figures that should be addressed before publication.

After carefully revising this new version, I believe that this manuscript represents a significant improvement over the first draft. In addition, the authors were able to provide clarification regarding the experimental design and also included R code.

Reviewer(s)' Comments to Author:

Referee: 2

Comments to the Author(s).

I appreciate how much work the authors have put into revising the manuscript, especially the addition of the code files, supplementary text, and new supplemental figures (S6 and S7). I think in particular they have done a solid job both in the text and response to reviewers arguing that their results are not due to technical issues, and better explaining their study design. I also appreciate their thorough responses to my critiques about statistical approaches and sequencing platforms. I only have a few minor comments.

- o Figure 1. It would be useful to add a sentence explaining that the additive log ratio uses Acetobacteraceae as the denominator, hence its value as 1 and other taxa are scaled relative to Acetobacteraceae. This addition would make the figure easier to interpret without reading the main text.
- o Figure 3 needs a legend.
- o The supplementary text states 'these cannot be from sequencing platforms', but I suggest rephrasing that 'these are highly unlikely to be from sequencing platforms'.

Author's Response to Decision Letter for (RSPB-2021-2193.R0)

See Appendix B.

Decision letter (RSPB-2021-2193.R1)

22-Nov-2021

Dear Dr Mazzucco

I am pleased to inform you that your manuscript entitled "Long-term gut microbiome dynamics in *Drosophila melanogaster* reveal environment-specific associations between bacterial taxa at the family level" has been accepted for publication in Proceedings B.

Data Accessibility section

Open Access

Paper charges

Sincerely,
Proceedings B
mailto: proceedingsb@royalsociety.org

Appendix A

Dear Editors,

We are delighted to resubmit our manuscript “Long-term gut microbiome dynamics in *Drosophila melanogaster* reveal environment-specific associations between specific families” (RSPB-2021-1203) for publication in *Proceedings B*. As you will see from our response below, we have addressed almost all comments of the reviewers. We hope that our manuscript will now be suitable for publication in *Proceedings B*.

Yours faithfully

Rupert Mazzucco & Christian Schlötterer

Associate Editor

Comments to Author:

Both reviewers recognized the value of this paper as it comprises a robust longitudinal dataset of microbiome changes in Drosophila. However, the reviewers also pointed out some inconsistencies related to sample sizes, suggested new figures to better explain the sampling design, and highlighted potentially confounding factors in the sequencing approach.

We are happy that both reviewers recognized the value of our study. To address the reviewer concerns we have reworked the presentation, modified and streamlined the text, and substantially expanded the SI. Please find out detailed response to all reviewer comments below.

Reviewer(s)' Comments to Author:

Referee: 1

Comments to the Author(s)

Mazzucco and Schlöterer present an analysis of microbiome composition in evolving populations of D. melanogaster over a replicated, long term experiment with hot and cold treatments. They find consistent dynamics of the microbiome at the family level across experimental replicates, suggesting metabolic consistency in the microbiome adaptation to the new environment. Furthermore, they find correlations between family level abundances that are consistent across replicates. Overall their results suggest a coarse-grained predictability of microbiome composition in adaptation to novel environments.

I find the work interesting, the analysis complete, and the conclusions for the most part justified. I have only minor comments that I think should be addressed before acceptance of the manuscript.

Main comment: while the analysis is technically solid, there are caveats to the microbiome sequencing and the associations are only correlations. For that reason, I think the conclusions as expressed in the title, abstract, and conclusion sections need to be softened.

Done.

Title: line 2, I would remove 'tight' from the title. While some correlations appear tight, they are just correlations, and overall the associations are not what I would call tight from looking at the data. Furthermore it is unclear what these associations might be caused by -- they may be completely indirect. A more justifiable title in my opinion could be 'Long-term gut microbiome dynamics in Drosophila melanogaster reveal environment-specific correlations between bacterial taxa at the family level'.

Done.

Abstract lines 22-23: I disagree that the evidence is 'unprecedented' or 'striking.' That wording should be toned down.

Done.

Introduction

line 55-57: the authors say that there are too many combinations for testing all configurations of Drosophila microbiome species, but see Lee 2019; Gould 2018. Both used fully factorial experimental designs with diversity similar to the present work to analyze Drosophila microbiomes. I suggest modifying the sentence to reflect this work or removing the sentence.

We have modified the sentence to read “testing the number of possible combinations is challenging“.

Methods

DNA extraction to determine microbial community has been shown to be a major source of error and variation in microbiome sequencing experiments. In particular, the extraction methods produce differential lysis of different types of bacteria, complicating the assignment of relative taxonomic information to different samples in different batches. Over a decade of trial and error by the microbiome community has yielded mechanical lysis as the standard approach. It appears that a standard genomic extraction without mechanical lysis was used. The trends seen in this dataset could thus be influenced by the extraction method. More explanation of the extraction methods and the caveats would resolve this issue.

We fully agree with the reviewer, but our Table S1 may be misleading in that it describes the full dataset, which also includes base populations, for which in some replicates a different DNA extraction method was used. However, base populations were not used in our analysis, so for all samples that were used the same (“High Salt”) DNA extraction method was applied. We have modified the sentence in the Methods section to clarify this point. As it is difficult to predict if this DNA extraction method discriminates some components of the microbiome, we did not expand this discussion, as the same discrimination would have been applied to all samples.

The age in days after eclosion when the flies were harvested for sequencing is also a potential influence on the bacterial community. According to Blum et al 2013, the fly gut is not substantially colonized until 4 days post eclosion. Changes in colonization occur as the fly ages. Because flies were sampled from 4 to 8 days post eclosion, significant ecological differences in colonization could occur over the initial

4 day period, leading to differences based on sampling. I think the authors should note this caveat.

Many thanks for pointing this out. We have added this caveat to the end of the Discussion subsection: Parallel dynamics across replicates.

The authors mention that nutrients influence the microbiome, but what are the nutrients in the fly media used?

We have now included the ingredients in the text.

What were the temperatures for the hot and cold environments? This information may be in a previous publication, but please provide it here.

The temperature regimes (10/20° and 18/28°C) are described at the beginning of the Methods section.

A detailed explanation of the culture conditions, including the approximate population sizes, food composition, how frequently the food is changed, etc would help in understanding why seemingly non-fly bacteria persist at high relative abundance in the experiment. For example, Bradyrhizobia and Burkholderia genomes are presumably environmental bacteria living on the food and not Drosophila gut residents. Explaining, based on the sampling methods, how sequencing reads from these strains could be observed at moderate abundances would greatly improve the manuscript.

We have added the population size in each replicate (~1000 flies in 5 bottles) to the Methods text. The food is never changed for a single fly generation: the eclosed flies are moved to fresh bottles where they lay eggs on a fresh nutrient medium, where the larvae stay until they pupate and eclose. Adults were frozen after 4–8 days. We have tried to make this more explicit in the text.

How many reads were bacterial and informative for each replicate? This could be added as column S in the table S1.

We have now added a table to the main text (Table 1) that gives read pair counts for the dominant families. We now also provide the Kraken reports in the SI that contain the full information for all sample runs.

line 277-284. This section is unclear (though the writing in the rest of the paper is clear and logical). Is the idea that this analysis is forthcoming in future work? Examining changes in DNA sequence rather than changes in abundance would not be affected by the DNA extraction methods and would provide an interesting perspective on microbiome evolution.

As an aside, the Katie Pollard's lab has developed methods for such inference in human microbiomes, allowing individual strains to be tracked over time. Might those methods be applied here [in future work]?

It was meant to mention possibilities for future research, but the Results section is maybe not the place for that, so we have removed that paragraph.

Discussion

How does the presence of Wolbachia correlate with the external microbiome analyzed here? Various authors have reported that correlations exist or not. Based on the authors' previous work with Wolbachia on the same data set, I think the Wolbachia dynamics should be related to the present analysis. For instance, could the trends in Clade V account for the shifts in abundance around generation 60?

The reviewer raises an interesting point. We had already analyzed the dynamics of Wolbachia in a previous publication (Mazzucco et al. (2020)), where we showed that different Wolbachia genotypes were present at the beginning of the experiment, but in both hot and cold conditions a single Wolbachia genotype predominated at the end. Given this complex dynamics of Wolbachia, we felt that accounting for Wolbachia diversity would result in an overly complex analysis and grouping it into a single genotype would also not do justice to the data. Hence, we decided to remove Wolbachia from this manuscript. Nevertheless, we would like to mention that we do not see much evidence for the existence of intermediate peaks in that data, which, however, appears to be a feature of the microbiome dynamics in the present study. We have added a sentence about this to the Discussion section “The microbiome turnover: driver or passenger?”.

Since examining bacterial reads from fly sequencing experiments might be repeated by future data mining efforts, some commentary would be helpful on the caveats of the methods and how these can be mitigated.

We cover all potential issues that occurred to us in the discussion and think that this could be helpful for future research.

Referee: 2

Comments to the Author(s)

*The goal of this work was to use a multigenerational host dataset to examine microbiome changes over a decade in two environmental conditions. Understanding longitudinal change is an important question in the microbiome literature, and few studies have long-term datasets to address it. This study uses up to 180 generations of Drosophila, with microbiome sampling of 7 to 16 of the generations. The results, that microbiome diversity and composition change over time and display different patterns in the two temperature regimes, have the potential to be a useful contribution to the microbiome literature. However, I have a few major concerns. I was asked to be a statistical reviewer, so most of my comments will focus on the data analysis. First, there are major clarity issues in what data were actually analyzed. Sample sizes are not included in the main text. The main text mentions that 100 generations of flies were used from the cold environment and 180 from the hot environment, with 5 replicate populations per time point “obtained from multiple timepoints” (L113). However, supplementary table S1 shows that 10-16 time points per replicate had microbiome data in the hot environment and 7-13 in the cold environment. The main text and figures need to state how many generations were actually *sampled* rather than only how many generations were run; a figure showing sampling design could be useful.*

We now mention the number of samples in the Abstract and at the beginning of the Discussion. We also added the information to the caption of Fig. 2; Fig. 4 already indicates this by showing samples as points). A figure showing the sampling setup is provided in the SI (Fig. S2), and we feel that is also where it should stay, since this was not really a design in the more narrow sense.

Similarly, it is unclear what results are shown in the figures. For example, the description of Fig. 1 in the text (L227-228) is, “the microbiomes of some consecutive time points were similar and clustered together (Fig. 1)” but the legend for Fig. 1 does not include the generation number or unit of distance on the Y axis so it’s not clear how this shows temporal generation differences, or even what generations or replicates are included in the figure. The title for Fig. 2 is “Figure 2 Microbial diversity depends on the environment and changes over time.” but nowhere in the legend text or figures is time mentioned. It is unclear what the points represent in figures S4, Fig. 4 includes colors in the scale not present on the figure, and Fig. S1 needs a legend for the color scheme.

We understand the concern of the reviewer that it may have been difficult to follow the presentation in the previous version of the manuscript. To streamline the presentation we have now removed the clustering of samples according to generation and the diversity changes among different clusters. The ancillary color scheme in Fig. S1 is now explained in the caption.

The second major issue is how the data were processed. The variation in microbiome results across DNA extraction methods and sequencing technologies is a well-known issue in the microbiome field, and samples need to be processed using consistent methods to be comparable. Because this study combined so many methodologic techniques and these techniques change over time (as the legend for Fig. S6, “Since samples were typically sequenced as they became available, the use of different platforms is correlated with time – which could affect the interpretation of the observed temporal dynamics of the microbiome (Fig. 3) if the sequencing technology affects species representation.”) it is hard to argue that temporal results represent true changes in the microbiome rather than data processing artifacts. For example, is the unique peak at generation 50 from using a distinct sequencing technology? The paper does a good job acknowledging multi-platform issues (L339 - 347) but it is not clear they controlled for these issues statistically. I do not find figures S6 and S7 sufficiently convincing that the results can be explained by time and not technology.

We understand these concerns, but since this is essentially a data-mining effort on an existing data set, there is not much we can now do about sample processing. As recognized by the reviewer, we acknowledge these issues and try to address them. We would like to point out that Figs. S6 and S7 (now Figs. S4 and S5) were never meant to show that results do not depend on technology, but were rather included to illustrate sources of variation. Fig. S3, on the other hand, was included to demonstrate that despite these sources of variation we identified real abundance changes. We provide two examples that cover different evidence. i) we show that similar peaks of abundance were picked up by different sequencing platforms, and ii) we show that the same sequencing platform uncovers a pronounced frequency change; this rules out that the abundance change is an artifact caused by different sequencing platforms. Many more examples can be found. To emphasize this we have now included new

Figs. S6 and S7 which provide in essence the same line of evidence: the same abundance pattern which coincides with a change in sequencing platform in one replicate occurs also in another replicate without a change of sequencing platform.

In fact, the argument can be made for all families and sequencing platforms as we discuss below:

- Acetobacteraceae show the same intermediate decrease in r06–r10 for all points covered by HiSeq 2000 or XTEN, making it clear that the decrease and subsequent recovery cannot be an artifact. The second time points in r08–r10 (GAllx) shows a higher fraction of Acetobacteraceae, but the difference in magnitude (very small in r09 vs, much larger in r08 and r10), as well as comparison with r01, where the third time point (GAllx) shows substantial decrease of Acetobacteraceae vs time points 1 and 2 (HiSeq 2000), makes it clear that GAllx does not systematically overestimate these bacteria vs. HiSeq 2000. Time points sequenced on HiSeq 2500 (for r01–r05) show both high and low fractions and do not stand out. This makes it seem unlikely that Acetobacteraceae dynamics are an artifact of sequencing platform.
- Lactobacillaceae dynamics look similar in r01–r05, despite the differences in sequencing platforms. The one visual outlier (r02, generation 50) does not correspond to a change in sequencing platform. While the lower fractions in generation 180 were all sequenced on XTEN, replicates r08 and r10 have pronounced peaks (generation 77) also sequenced on XTEN. The second time points in r08–r10 (GAllx) do not stand out in any way from the surrounding points sequenced on HiSeq 2000. Again it seems very unlikely that Lactobacillaceae dynamics are an artifact of sequencing platform.
- Streptomyetaceae occur at higher fractions in earlier generations in r02 (all sequenced on HiSeq 2000), but not r03 (all sequenced on HiSeq 2000). The time points sequenced on GAllx do not stand out from points sequenced on HiSeq 2000 (e.g., r03–r05), nor do the time points sequenced on HiSeq 2500 or XTEN. The pronounced increase in r02 occurs over generations fully covered by HiSeq 2000. As before, it does not seem that Streptomyetaceae dynamics are due to changes in sequencing platform.
- As for Enterobacteriaceae, r06, where also the early generations were sequenced on XTEN, also shows the late increase of Enterobacteriaceae around the same time it occurs in r08 and r09, where it coincides with the switch from HiSeq 2000 to XTEN; , similarly in the hot environment. The visual outlier in r02 at generation 89 was sequenced on HiSeq 2500 as were its neighboring time points, which do not stand out. Again it follows that the Enterobacteriaceae dynamics cannot be an artifact of the platform change.
- The pronounced Burkholderiaceae peak in r08–r10 at first glance seems to coincide with the change to and from the HiSeq 2000 platform, but there is the first time point also sequenced on HiSeq 2000, which does not show increased abundance. Also, the pronounced peak is missing in replicate r07, where all early generations were sequenced on HiSeq 2000, and which in fact looks more similar to r06, where most early generations were sequenced on XTEN. There is a high parallelism in replicates r04 and r05, but its most distinctive pattern represented by the three time points between generations 50 and 100 is fully covered by HiSeq 2000. Again we must reject the idea that Burkholderiaceae dynamics are an artifact of sequencing platform.

- The Leuconostocaceae peak occurs in r06 pretty much in the same way as in r08, r09, and r10, although covered by HiSeq XTEN instead of HiSeq 2000. It follows immediately that the Leuconostocaceae peak in the cold environment cannot be an artifact from switching the sequencing platform to XTEN in the later generations. In the hot environment, we see pretty much the same pattern in all replicates despite the sequencing differences, and the only visual outlier (first time point in r03) was sequenced on HiSeq 2000 like all following generations < 100. It does not seem that Leuconostocaceae dynamics are an artifact of the sequencing platform.
- Finally, also the few Bradyrhizobiaceae peaks across replicates (generation 33 in r08–r10 and some early time point in r01–r03) are fully covered by time point sequenced on the HiSeq 2000 platform, and time points sequenced on other platforms do not stand out anywhere. Also the Bradyrhizobiaceae dynamics thus do not seem to be artifact of the sequencing platform.

Since we felt that this discussion will overwhelm the reader, we limit the discussion included in SI.C to the two most obvious examples.

We would like to emphasize that we stick to a verbal discussion because there are simply not enough data points to add, e.g., sequencing platform as a factor to the model and consider its interaction with generation.

The third issue is clarity in the statistical results. Piecewise linear models are not commonly used in microbiome studies, so a sentence justifying their use would be helpful.

We have added such a sentence to the respective Methods subsection.

It also would be useful to list what variables were included in the models- e.g. were replicate, DNA extraction method, and sequencing platform controlled for? The statistical results are often reported in qualitative terms, with no test statistics or other numbers shown. For example, L168 states that “analyses at the genus level provided a very similar picture” but these results are not shown; L231 – 233 describes values as ‘statistically more diverse’ with no stats; L242-243 lists the most abundant families but not their percentage abundances; and L270-L273 describes values ‘highly variable between timepoints’. Is there supplementary info that I’m missing that includes the actual statistical model results? The SI info I have only shows figures.

Thank you for these points. We have now added a Table 1 that also shows the genera that the families comprise; Table 1 and the new Fig. 1 also show the overall abundance of the dominant families both by read pair count and fraction; Kraken reports, Bracken output, and the actual statistical model results are now provided as SI materials, as is the R code used for the analysis and plots. The paragraph containing the referenced L231–L233 has been removed entirely. The DNA extraction method was the same for all samples considered in the analysis; as for the platform issue, please refer to our answer above. SI.C now has models fitted by replicate and some discussion of them.

There are also a few minor points that would stand to improve the paper, bulleted below:

o L172-L177 Please add a reference for using estimated instead of observed fractions.

We have now added some references.

o L205 - L217- This paragraph should be removed or altered- it reads like an abstract rather than as part of a results section.

We have reduced the paragraph to a short introduction of the actual results.

o Fig. 4- Is there a quantitative analysis on the 'obvious groups'?

The obvious groups have been removed from the text.

o Fig. 5- The grey frames are very hard to see.

We have now modified the design of this figure (now Fig. 4) to improve visibility of the significant models.

Appendix B

Dear Editors,

We are happy to read that the reviewer appreciated our efforts to revise the manuscript. This resubmission of our manuscript “Long-term gut microbiome dynamics in *Drosophila melanogaster* reveal environment-specific associations between specific families” (RSPB-2021-1203) accounts for the few remaining comments of the reviewer.

Yours faithfully

Rupert Mazzucco & Christian Schlötterer

Referee: 2

Comments to the Author(s).

I appreciate how much work the authors have put into revising the manuscript, especially the addition of the code files, supplementary text, and new supplemental figures (S6 and S7). I think in particular they have done a solid job both in the text and response to reviewers arguing that their results are not due to technical issues, and better explaining their study design. I also appreciate their thorough responses to my critiques about statistical approaches and sequencing platforms. I only have a few minor comments.

Many thanks for this friendly assessment and your helpful comments. We have addressed them as shown below.

o Figure 1. It would be useful to add a sentence explaining that the additive log ratio uses Acetobacteraceae as the denominator, hence its value as 1 and other taxa are scaled relative to Acetobacteraceae. This addition would make the figure easier to interpret without reading the main text.

We have added such an explanation to the figure caption.

o Figure 3 needs a legend.

We have made sure that Fig. 3 has a legend and a caption.

o The supplementary text states ‘these cannot be from sequencing platforms’, but I suggest rephrasing that ‘these are highly unlikely to be from sequencing platforms’.

We have rephrased the respective sentences in the SI accordingly.